# Infective endocarditis post-transcatheter aortic valve implantation (TAVI), microbiological profile and clinical outcomes: A systematic review

Adnan Khan[1]*, Aqsa Aslam[1], Khawar Naeem Satti[2], Sana Ashiq[1]

**1** Sharif Medical and Dental College, Lahore, Pakistan, **2** Senior Registrar Rawalpindi Institute of Cardiology, Rawalpindi, Pakistan

* adnan.khan@outlook.com

## Abstract

### Background

The data on infective endocarditis after transcatheter aortic valve implantation (TAVI) is scarce and limited to case reports and case series in the literature. It is the need of the hour to analyze the available data on post-TAVI infective endocarditis from the available literature. The objectives of this systematic review were to evaluate the incidence of infective endocarditis after transcatheter aortic valve implantation, its microbiological profile and clinical outcomes. It will help us to improve the antibiotic prophylaxis strategies and treatment options for infective endocarditis in the context of TAVI.

### Methods

EMBASE, Medline and the CENTRAL trials registry of the Cochrane Collaboration were searched for articles on infective endocarditis in post-TAVI patients till October 2018. Eleven articles were included in the systematic review. The outcomes assessed werethe incidence of infective endocarditis, its microbiological profile andclinical outcomes including major adverse cardiac event (MACE), net adverse clinical event (NACE), surgical intervention and valve-in-valve procedure.

### Results

The incidence of infective endocarditis varied from 0%-14.3% in the included studies, the mean was3.25%. The average duration of follow-up was 474 days (1.3 years). *Enterococci* were the most common causative organism isolated from 25.9% of cases followed by *Staphylococcus aureus* (16.1%) and coagulase-negative *Staphylococcus* species (14.7%). The mean in-hospital mortality and mortality at follow-up was 29.5% and 29.9%, respectively. The cumulative incidence of heart failure, stroke and major bleeding were 37.1%, 5.3% and 11.3%,respectively. Only a single study by Martinez-Selles et al. reported arrhythmias in 20% cases. The septic shock occurred in 10% and 27.7% post-TAVI infective

**Data Availability Statement:** All relevant data are within the manuscript and its Supporting Information files.

**Funding:** The authors received no specific funding for this work.

**Competing interests:** The authors have declared that no competing interests exist.

endocarditis patients according to 2 studies. The surgical intervention and valve-in-valve procedure were reported in 11.4% and 6.4% cases, respectively.

## Conclusion

The incidence of post-TAVI infective endocarditis is low being 3.25% but it is associated with high mortality and complications. The most common complication is heart failure with a cumulative incidence of 37.1%. *Enterococci* are the most common causative organism isolated from 25.9% of cases followed by *Staphylococcus aureus* in 16.1% of cases. Appropriate measures should be taken to prevent infective endocarditis in post-TAVI patients including adequate antibiotics prophylaxis directed specifically against these organisms.

## Study registration

PROSPERO registration number CRD42018115943.

## Introduction

Infective endocarditis (IE) is an uncommon infectious disease but with significant mortality and morbidity [1]. The mortality rate of infective endocarditis is 25% [1]. In most population surveys, its incidence ranges from 3–7 per 100,000 people per year [2]. Infective endocarditis ranks fourth among the life-threatening infections. In 2010, it was estimated that IE causes 1.58 million disability-adjusted life-years worldwide [3]. The causative organism isolated is *Viridans streptococci* in 35–45% of the patients [4]. According to a survey in France, the microbiological profile of infective endocarditis has changed in recent years [5]. *Staphylococcus aureus* causes the majority of the cases of infective endocarditis in the industrialized world. There is a higher incidence of infective endocarditis in older age, those with prosthetic heart valves and cardiac devices while at the same time, there is a decreased proportion of IE in rheumatic heart disease [6].

Infective endocarditis presents with high-grade fever, valvulitis, peripheral emboli, immunological phenomenon and sustained bacteremia or fungemia. However, the typical history and clinical manifestations are not present in most of the patients. So, the diagnosis of infective endocarditis relies on a highly sensitive and specific diagnostic strategy. In 1994, a diagnostic scheme was formulated in Duke University Medical Center. According to this scheme, patients with suspected infective endocarditis were allocated into three classes: definite, possible and rejected cases(Table 1) [7].

**Table 1. Definitions of definite, possible and rejected infective endocarditis [7].**

| | | |
|---|---|---|
| **Definite IE** | **Pathological criteria** | Microorganisms demonstrated by culture or histological examination of vegetation, vegetation that has embolized, or an intracardiac abscess specimen; or pathological lesions; vegetation or intracardiac abscess confirmed by histological examination showing active endocarditis |
| | **Clinical criteria** | 2 Major criteria, 1 major criterion and 3 minor criteria, or 5 minor criteria |
| **Possible IE** | | 1 Major criterion and 1 minor criterion, or 3 minor criteria |
| **Rejected** | | Firm alterative diagnosis explaining evidence of IE; or resolution of IE syndrome with antibiotic therapy for ≤ 4 d; or no pathological evidence of IE at surgery or autopsy with antibiotic therapy for ≤ 4 d; or does not meet criteria for possible IE as above |

Antibiotic prophylaxis is given for the prevention of infective endocarditis in high-risk persons before dental treatment [8]. The high-risk persons for infective endocarditis include those withtheprevious episode of infective endocarditis, congenital heart defects and prosthetic heart valves. Individuals with the history of rheumatic fever, heart murmur and native valve disease are at moderate risk of infective endocarditis[4].

Aortic stenosis is the most prevalent valvular abnormality in adults with a higherincidencein advanced age. The frequency of aortic stenosis has increased due to increasing life expectancies[9]. The patients with aortic stenosis are treated with conventional cardiac surgery, surgical aortic valve replacement (SAVR). Until recently due to significant mortality associated with SAVR, elderly patients with co-morbidities were not treated[10].

Transcatheter aortic valve implantation (TAVI) was introduced in 2006 as a revolutionary intervention for severe aortic stenosis. It is a less invasive procedure for high-risk patients or patients who are inoperable by SAVR[11, 12].Sternotomy is not required in TAVI. A bioprosthetic valve is implanted over the native valve using a catheter. The mortality rate after TAVI is reported to be 14%-31% after 1 year[13]. The commonly used TAVI systems are the Edwards SAPIEN valve and the CoreValve®. Both of them are effective and safe[14]. Transcatheter aortic valve implantation can be performed through various approaches such as transfemoral, transapical, subclavian or direct aortic approach. The transfemoral approach is done in 80–90% of the cases and is appropriate for both types of the valve[15]. The success rate after TAVI is more than 90% and 30-day procedural mortality rates less than 10%[16]. The majority of the complications after TAVI are technical and device-related including conduction disturbances and periprosthetic paravalvular leaks[17].After the success of TAVI in aortic stenosis, several strategies have been established to treat aortic regurgitation with TAVI [18].

Infective endocarditis (IE) is a fatal complication of TAVI[19]. The incidence of infective endocarditis after TAVI is low, although it is a major cause of heart failure. As TAVI is becoming more popular with time, the magnitude of post-TAVI infective endocarditis will rise[20]. The patients with post-TAVI infective endocarditis have an atypical clinical presentation causing a delay its diagnosis and treatment[21].Surgical aortic valve replacement is often required to explant the valve with endocarditis and most of these patients are inoperable or at high risk for SAVR[12]. The majority of the patients who underwent TAVI are elderly and infective endocarditis could have the worst prognosis in them[22].The leaflets of transcatheter valve prostheses contain a greater quantity of metal in the stent frame in contrast to the surgical valves. This factor may change the outcome and management of IE[23].

There is limited data on infective endocarditis after TAVI, its microbiological profile, clinical outcomes and treatment modalities. Most of the data is limited to case reports and small series, which can lead to publication bias[19]. In the prospective randomized Placement of Aortic Transcatheter Valves (PARTNER) trial, the incidence of infective endocarditis after TAVI is reported from 0.1% to 3.03%, with no difference between TAVI and SAVR[24,25]. According to other studies, infective endocarditis occurs in 0.5% to 3.1% of patients after TAVI[26,27]. Similar incidence rates of infective endocarditis are reported after surgical valve replacement[28].

Fig 1 shows the transesophageal (long and short-axis) view and fluoro-deoxyglucose positron emission tomography of the post-TAVI endocarditis and Fig 2 illustrates theprocedure of TAVI.

The clinical picture of post-TAVI IE varies from nonspecific symptoms to acute infection or sepsis with fever, heart failure or embolic stroke. As most of these patients are elderly, they frequently present with atypical symptoms and signs. Greater than 50% of the patients present with heart failure and 20% have non-specific symptoms e.g. malaise, weakness or weight loss. High-grade fever and heart murmur are relatively less common in post-TAVI IE as compared

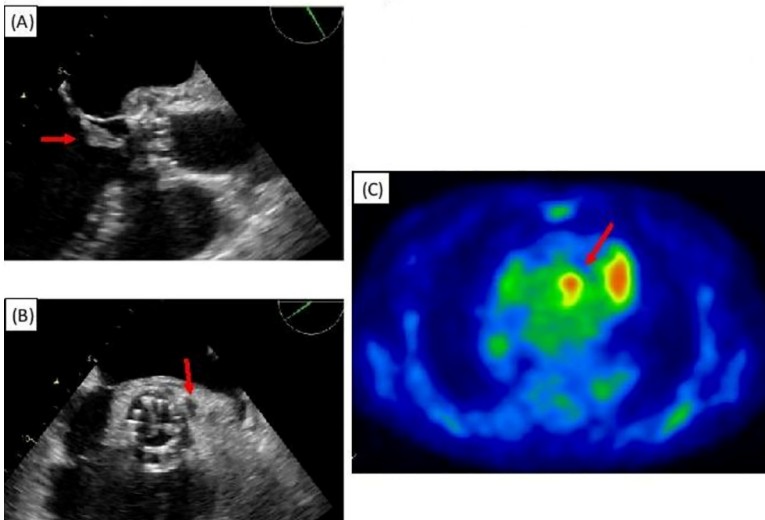

**Fig 1. "Transcatheter heart valve (THV) endocarditis.** A. Long-axis transesophageal view showing typical vegetation attached to the ventricular side of a THV (red arrow). B. Short-axis transesophageal view showing de novo peri-prosthetic echo-free cavities and thickened areas (red arrow). C. 4 fluoro-deoxyglucose positron emission tomography (FDG-PET) depicting cells with an enhanced glucose metabolism at the level of THV, thus corroborating the diagnosis of endocarditis" [29].

to native valve IE. The diagnosis of post-TAVI IE by echocardiography is more difficult than native valve IE due to differences in the technique of valve implantation. The detection of small vegetation in post-TAVI IE is limited as the valve contains large amounts of metal which create reflectance and a shadowing effect [30]. According to guidelines, early surgery is indicated in complicated cases of post-TAVI IE. Because these patients are high-risk, surgery is contraindicated in most of the patients with post-TAVI IE [31].

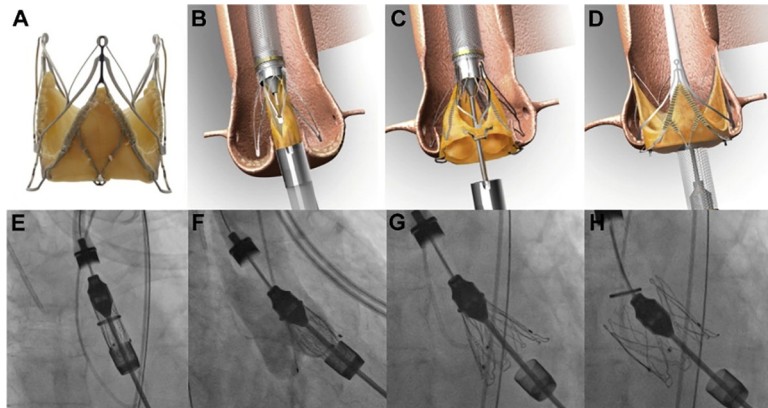

**Fig 2.** "The JenaValve transcatheter heart valve (THV) prosthesis (JenaValve Technology GmbH, Munich, Germany), a trileaflet porcine root tissue valve attached to a nitinol stent (A) and its implantation in illustration (B to D) and fluoroscopy (E to H). Release of the positioning feelers and placement into the aortic sinuses enables anatomic orientation (B and F). After correct orientation has been verified in 2 different fluoroscopic angulations, release of the lower stent part facilitates the clipping of the native aortic valve leaflets to the device and expansion of the stent allowing for secure anchoring even in the absence of valve calcium (C and G). Release of the upper stent part completes deployment of the valve prosthesis (D and H)" [18].

### Rationale

Infective endocarditis is a rare but fatal complication of TAVI with its incidence ranging from 0.1%-3.03%. It has a high mortality rate with many complications [1,23].Transcatheter aortic valve implantation is gaining popularity day by day because it is feasible, less invasive technique and the only treatment option in patients at high surgical risk for severe symptomatic aortic stenosis [11]. Due to this, the incidence of post-TAVI infective endocarditis is expected to rise in the coming years. The data on infective endocarditis after TAVI is scarce and limited to case reports and case series in the literature [19,20]. It is the need of the hour to analyze the available data on post-TAVI infective endocarditis from the available literature. This study wasdesigned to conduct a systematic review of the incidence of infective endocarditis after TAVI, the causative pathogens isolated from these patients and the clinical outcomes. It will help us to improve the antibiotic prophylaxis strategies and treatment options for infective endocarditis in the context of TAVI.

### Objectives

The objectives of this systematic review were to evaluate the incidence of infective endocarditis after TAVI, its microbiological profile and clinical outcomes.

### Material and methods

The study was done according to the Preferred Reporting Items for Systematic Reviews and Meta-analyses (PRISMA) guidelines and is registered with PROSPERO International Prospective Register of Systematic Reviews (PROSPERO registration number CRD42018115943).

### Eligibility criteria

The following criteria were used for the selection of studies:

### Study designs

The retrospective, prospective and observational studies were eligible for the systematic review. The case reports and case series were excluded.

### Participants

The studies included adult humans of either gender with age $\geq$70 years who underwent TAVI. The minimum follow-up time was 6 months.

### Interventions

The intervention was TAVI. Healthcare provision and follow-up of the patients after TAVI to look for the incidence of infective endocarditis & its clinical outcomes. Investigating the causative organisms of infective endocarditis.

"Transcatheter heart valve endocarditis is defined following Duke's modified criteria or evidence of abscess/paravalvular leak/pus/vegetation confirmed as secondary to infection by histological or bacteriological studies or evidence of abnormal tracer uptake around the site of the prosthetic valve by F-fluoro-deoxyglucose positron emission tomography"[29].

### Comparators

There was no control group for comparison.

## Outcomes

The primary outcome was:

➢ Incidence of post-TAVI infective endocarditis

The secondary outcomes were:

➢ Microbiological profile of infective endocarditis

➢ MACE including in-hospital mortality & mortality at follow-up, heart failure, stroke, major bleeding and arrhythmias

➢ NACE including septic shock

➢ Surgical intervention

➢ Valve-in-valve procedure

"Sepsis is defined as an infection that triggers a particular Systemic Inflammatory Response Syndrome (SIRS). This is characterized by body temperature outside 36˚C—38˚C, HR >90 beats/min, respiratory rate >20/min, WBC count >12,000/mm3 or <4,000/mm3. Patients with infections plus two or more elements of the SIRS meet the criteria for sepsis. Those who have end organ failure are considered as having severe sepsis; and those who have refractory hypotension along with the above said criteria are considered to be in septic shock [32]."

## Timing

All the included studies had a follow-up time of at least 6 months after TAVI.

## Setting

The study conducted in any type of setting was included.

## Language

Studies reported in the English language only.

## Information sources

All the relevant articles in English with text words related to infective endocarditis and transcatheter aortic valve implantation (TAVI) were searched in MEDLINE (PubMed), EMBASE (OVID interface) and the Cochrane Central Register of Controlled Trials (Wiley interface) till October 2018. The literature search was limited to human subjects. The case reports and case series were excluded.

## Search strategy

Medline, EMBASE and the CENTRAL trials registry of the Cochrane Collaboration were searched for keywords, including 'Transcatheter aortic valve implantation', 'Transcatheter aortic valve replacement', 'TAVI', 'TAVR', 'Endocarditis', 'Infective endocarditis', 'Prosthetic valve endocarditis', 'Infective endocarditis after TAVI', 'Incidence and clinical impact of infective endocarditis on TAVI', 'TAVI-associated infective endocarditis', 'Prosthetic valve endocarditis after transcatheter valve replacement', 'Causative organisms of post-TAVI infective endocarditis', 'Clinical outcomes of infective endocarditis after TAVI', 'In-hospital mortality', 'Mortality at follow-up', 'Transcatheter heart failure' and 'Outcomes of TAVI'.

## Data management, selection process and data collection

The first and second authorswere the principal investigators. Each author individually read all the relevant articles. The articles meeting the eligibility criteria were included for the systematic review. The search results from each database were saved in EndNote X9 and duplicates were removed.

## Data items

Data was recorded and tabulated including author name, year of publication, sample size, study type, follow-up time, mean age & gender of patients, primary and secondary outcomes. It ensured uniformity between the authors and integration of findings.

## Outcomes and prioritizations

The outcomes were the incidence of infective endocarditis after TAVI, its microbiological profile and clinical outcomes of infective endocarditis.

## Data synthesis

The search strategy shortlisted 137 articles, out of which 44 articles were relevant. Forty four articles on infective endocarditis in post-TAVI patients were assessed for full text. Out of these, 11 studies were included in the systematic review. All the studies were retrospective or observational with the follow-up duration of at least 6 months. Thirty three articles did not meet the inclusion criteria as these were case reports, case series and systematic reviews. The meta-analysis could not be done as there was no control group for comparison.The PRISMA flow diagram for the study protocol is shown in Fig 3.

## Data analysis

The data entry and analysis was done using the Statistical Package for the Social Sciences (SPSS) version 25. The follow-up duration and incidence of IE were evaluated as mean. The microbiological profile and clinical outcomes of post-TAVI IE were expressed as frequency and percentage.

**Risk of bias.** The Newcastle-Ottawa-Scale was used to calculate the risk of bias in the included studies as shown in Table 2.

## Results

A total of 44 articles on infective endocarditis in TAVI patients were assessed for full-text. Thirty three articles were excluded as these articles were case reports, case series and systematic reviews. Eleven articles were included in this systematic review with the average duration of follow-up 474 days (1.3 years). Table 3 shows the important parameters of the included studies.

Out of 11 included articles,the incidence of infective endocarditis varied from 0%-14.3%, the mean is 3.25%. This may be attributed to hospital, cardiologist and patient-related factors or the sample size. The incidence of post-TAVI infective endocarditis is shown in Fig 4.

The clinical outcomes of post-TAVI infective endocarditis are summarized in Table 5.

Out of 11 included studies, the data on antibiotic prophylaxis is detailed in 2 studies by Regueiro et al. and Amat-Santos et al. In a study by Regueiro et al., β lactam antibiotics were given in 195(78%) patients and vancomycin in 15(6%) patients for prophylaxis [28]. Amat-Santos et al. reported in his study that all patients with post-TAVI IE received prophylactic antibiotics

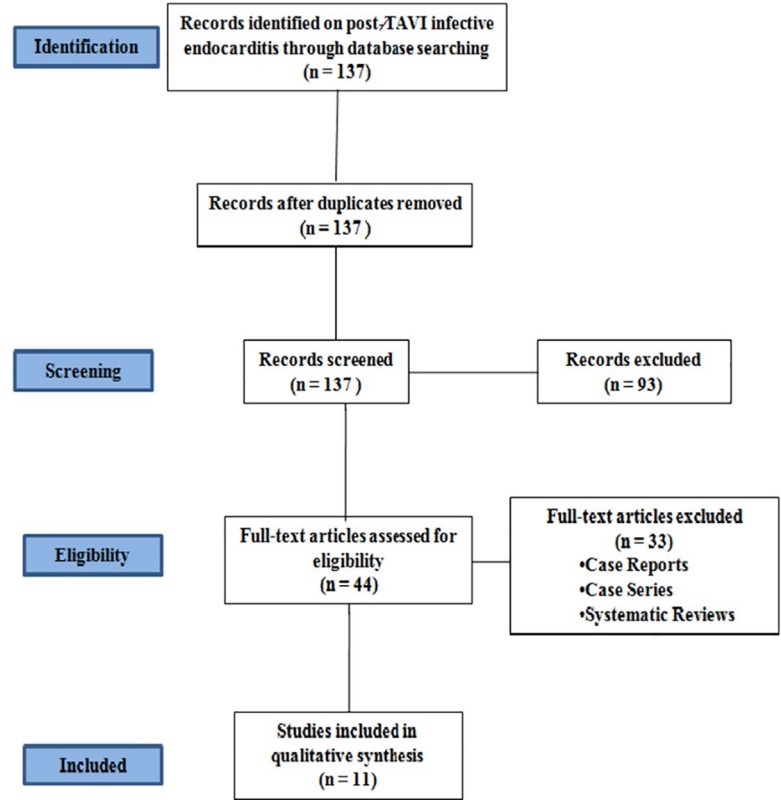

**Fig 3. PRISMA flow diagram.**

(n = 53). Cephalosporins, vancomycin and penicillin were given in 30(56.6%), 14(26.4%) and 9(17%) patients, respectively [23].

Latib et al. conducted a multicenter study in which he stated that all the patients were given prophylactic antibiotics depending on institutional practices. However, the details of these antibiotics are not mentioned in the article [32]. In a study by Olsen et al., patients received prophylactic antibiotics (amoxicillin or roxithromycin) only before undergoing invasive dental procedure [27].

Studies by Regueiroet al., Amat-Santos et al., Martinez-Selles et al., Puls et al. and Olsen et al. mentioned the antibiotics used for the treatment of post-TAVI IE. Regueiro et al. reported in his study that 164 post-TAVI IE patients were treated with β lactam antibiotics (only β lactam antibiotics in 38 patients and in combination with other antibiotics in 126 patients) and vancomycin was used in 53 patients [28].

In a study by Amat-Santos et al., all the patients with post-TAVI IE were treated with prolonged antibiotic therapy of 4 weeks. β lactam antibiotics, gentamicin and vancomycin were given in 21(39.6%), 20(37.7%) and 16(30.2%) patients. Seven patients with positive blood cultures for *Staphylococcusaureus* and *Staphylococcusepidermidis* received rifampicin. Five of the 6 patients with post-TAVI IE due to methicillin-resistant *Staphylococcusaureus* received daptomycin [23]. Some patients had received antibiotics in combination. That is why the percentage of antibiotics is not 100%.

The antibiotics used for treating post-TAVI IE in 18 patients in a study by Oslen et al. were vancomycin in combination with linezolid/rifampicin in 5 patients, penicillin or ampicillin or dicloxacillin in combination with gentamicin/fusidic acid/linezolid/rifampicin in 8 patients,

**Table 2. Risk of bias in individual studies.**

| Study/ Author | Selection | | | | Comparability | Exposure | | |
|---|---|---|---|---|---|---|---|---|
| | Case definition adequate | Representativeness of the cases | Selection of Controls | Definition of Controls | Comparability of cases and controls on the basis of the design or analysis | Ascertainment of exposure | Same method of ascertainment for cases and controls | Non-Response rate |
| Amat-Santos[23] | Low | Low | Low | Low | Low | Low | Low | Low |
| Puls[34] | Low | Low | Some concern | Some concern | Some concern | Low | Some concern | Low |
| Latib[33] | Low | Low | Some concern | Some concern | Some concern | Low | Some concern | Low |
| Olsen[27] | Low | Low | Low | Low | Low | Low | Low | Low |
| Martinez-Selles[22] | Low | Low | Low | Low | Low | Low | Low | Low |
| Regueiro [28] | Low | Low | Some concern | Some concern | Some concern | Low | Some concern | Low |
| Seiffert[18] | Low | Low | Some concern | Some concern | Some concern | Low | Some concern | Low |
| Spartera [29] | Low | Low | Some concern | Some concern | Some concern | Low | Some concern | Low |
| Kosek[36] | Low | Low | Some concern | Some concern | Some concern | Low | Some concern | Low |
| Gallouche [35] | Low | Low | Some concern | Some concern | Some concern | Low | Some concern | Low |
| Doss[37] | Low | Low | Some concern | Some concern | Some concern | Low | Some concern | Low |

cefuroxime or ceftriaxone in combination with fusidic acid/ciprofloxacin/rifampicin in 5 patients [27].

Puls et al. mentioned the following antibiotics used for treating 5 patients with post-TAVI IE: vancomycin & rifampicin alone or in combination with gentamicin in 2 patients, ampicillin with ciprofloxacin/gentamicin in 1 patient and ceftriaxone in 1 patient. One patient received parenteral antibiotics, the names of which are not mentioned in the study [34].

In a study by Martinez-Selles et al., out of 10 post-TAVI IE patients, 9 patients received β lactam antibiotics. Two out of these 9 patients were also given aminoglycosides. One patient had fungal endocarditis and was treated with fluconozole&caspofungin [22]. The antibiotics used for the treatment of post-TAVI IE in 5 studies are tabulated in Table 6.

## Discussion

The results of our study showed that the mean incidence of post-TAVI IE is 3.25%. The most common causative organism is *Enterococci* (25.9%) followed by *Staphylococcus aureus* (16.1%) and *coagulase-negative Staphylococcus* species (14.7%). Amat-Santos et al. also reported *Enterococci* (34.4%) as the most common organism causing IE after TAVI but the second common organism was *coagulase-negative Staphylococcus* species (18.7%) [26]. In contrast, in a systematic review by Eisen et al., *coagulase-negative Staphylococcus* species was the most common (30%) cause of post-TAVI IE followed by *Enterococci* (20%) [19].

In our study, the mean incidence of in-hospital mortality is 29.5% and mortality at follow-up is 29.9%. Similarly, the high mortality rate was reported in other systematic reviews. Post-TAVI IE was responsible for 34.4% and 40% mortality in studies by Amat-Santos et al. and Eisen et al.,respectively [26,19]. According to our study, surgical intervention and valve-in-valve procedure for the treatment of post-TAVI infective endocarditis were performed in

**Table 3. Characteristics of included studies.**

| Study/ Author | Year | Study Type | Journal | Sample size | Follow-up | Mean Age (years) All TAVI patients | Mean Age (years)THV-e patients | Gender All TAVI patients | GenderTHV-e patients |
|---|---|---|---|---|---|---|---|---|---|
| Amat-Santos [23] | 2015 | Retrospective | Circulation | 7944 | 1.1±1.2 years | 81±8 | 79±8 | | 23(43.4%) females30 (56.6%) males |
| Puls[34] | 2013 | Prospective | Eurointervention | 180 | 319 days | | 83.4 | | 3(60%) females2 (40%) males |
| Latib[33] | 2014 | Retrospective | J Am Coll Cardiol | 2572 | 393 days | | 80±6 | | |
| Olsen[27] | 2015 | Retrospective | Circ Cardiovasc Interv | 509 | 1.4 years | 80±6.9 | 78±6.9 | 213(42%) females296(58%) males | 1(6%) female17(94%) males |
| Martinez-Selles[22] | 2016 | Prospective | Eurointervention | 952 | 423 days | 79.5 | | | 4(40%) females6 (60%) males |
| Regueiro[28] | 2016 | Retrospective | JAMA | 20006 | 10.5 months | 81.8 | 78.9 | | 91(36.4%) females159(63.6%) males |
| Seiffert[18] | 2014 | Retrospective | J Am Coll CardiolIntv | 31 | 235 days | 73.8±9.1 | | 11(35.5%) females20(64.5%) males | |
| Spartera[29] | 2018 | Retrospective | Echocardiography | 621 | 402 days | 80±7.9 | 78±6.5 | 333(53.6%) females288(46.4%) males | 4(50%) females4 (50%) males |
| Kosek[36] | 2015 | Retrospective | Kardiol Pol | 7 | 12 months | 77.7 | 80 | 4(57%) females3 (43%) males | 1 male |
| Gallouche [35] | 2018 | Retrospective | J Hosp Infect | 326 | 460 days | 85 | 79.8 | 191(58.6%) females135(41.4%) males | 5(83.3%) females1 (16.7%) male |
| Doss[37] | 2012 | Prospective | Ann Thorac Surg | 100 | 3.8±2 years | 85±6 years | | 71(71%) females29 (29%) males | |

**TAVI:** Transcatheter aortic valve implantation, **THV-e:** Transcatheter heart valve endocarditis

11.4% and 6.4% of the cases, respectively. Amat-Santos et al. and Eisen et al.reported surgical intervention in 40% and 30% of the post-TAVI IE patients, respectively which is much higher than our results [26,19].

A systematic review was done on infective endocarditis after transcatheter aortic and pulmonary valve replacement. Twenty eight articles published between 2000 and 2013 including 16 on TAVI were analyzed with 32 post-TAVI infective endocarditis patients. The follow-up duration was 3 to 9 months. According to this review, *Enterococci* were the most common organism of IE after TAVI accounting for 11(34.4%) of the cases. Other organisms isolated were *coagulase-negative Staphylococcus* species in 6(18.7%), other *Streptococcus* species in 5 (15.6%), *Staphylococcus aureus* in 2(6.3%), gram negative rods in 2(6.3%), *Moraxella* in 1 (3.1%), *Candida albicans* in 1(3.1%), *Histoplasma* in 1(3.1%) and *Corynebacterium* in 1(3.1%). Two (6.3%) patients had negative blood cultures. The patients presented with fever, chills, anorexia, congestive heart failure, stroke, hemiparesis, sepsis and limb ischemia. Thirteen (41%) patients required surgical intervention and 11(34.4%) patients died due to infective endocarditis[26].

Another systematic review was conducted by Eisen et al. including 10 cases, 8 were from case reports and 2 cases were presented in congresses. The microorganisms isolated from blood cultures were *coagulase-negative Staphylococcus* species in 3(30%), *Enterococci* in 2 (20%), *Candida albicans,Moraxella*, *Corynebacterium*, other *Streptococcus* species and *Histoplasma* (1 in each) patients. Three (30%) patients underwent surgery and 4(40%) patients died

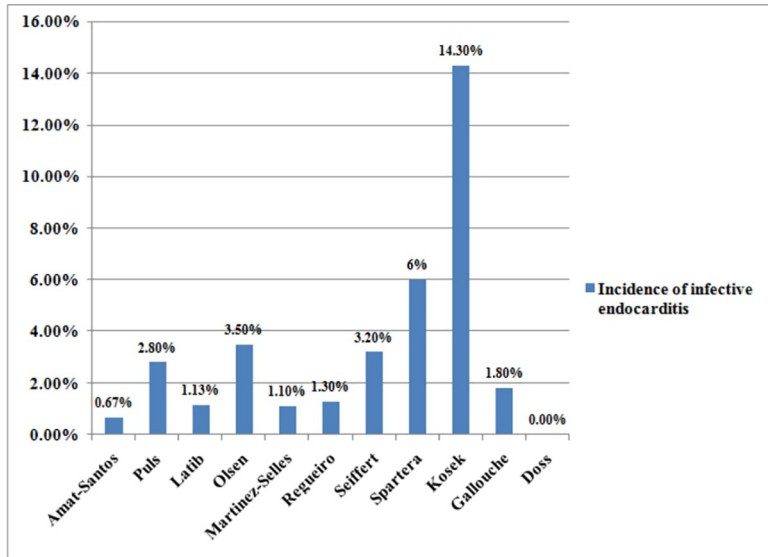

**Fig 4. Percentage of post-TAVI infective endocarditis in studies included in the systematic review.** The microbiological profile of post-TAVI infective endocarditis is reported in 8 studies. *Enterococci* are the most common causative organism isolated from 25.9% of cases followed by *Staphylococcus aureus* (16.1%), *coagulase-negative Staphylococcus* species (14.7%), other *Streptococcus* species (12.5%), *Viridans streptococci* (8.5%), gram negative rods/ *HACEK*/*candida* species (11%) and polymicrobial (0.42%).The cultures were negative in 6.7% of cases. The causative organisms of infective endocarditis are shown in Fig 5.

**Table 4. Incidence of post-TAVI infective endocarditis in the included studies with its causative organisms.**

| Amat-Santos | Puls | Latib | Olsen | Martinez-Selles | Regueiro | Seiffert | Spartera | Kosek | Gallouche | Doss |
|---|---|---|---|---|---|---|---|---|---|---|
| **Incidence of Infective Endocarditic** | | | | | | | | | | |
| 53(0.67%) | 5(2.8%) | 29(1.13%) | 18(3.5%) | 10(1.1%) | 250(1.3%) | 1 (3.2%) | 8(6%) | 1 (14.3%) | 6(1.8%) | 0 |
| **Causative Organisms** | | | | | | | | | | |
| *CoNS* 13 (24.5%) | *Enterococci*2 (40%) | *Enterococci*5 (17.24%) | *Enterococci* 6 (33%) | *Enterococci* 3 (30%) | *Enterococci* 57(24.6%) | | *Enterococci* 2 (25%) | | *Staph aureus* 1 (16.67%) | |
| *Staph aureus* 11(20.75%) | *Staph aureus* 1 (20%) | CoNS 4(13.8%) | *Staph aureus* 4 (22%) | *Other Streptococc*2 (20%) | *Staph aureus* 54(23.3%) | | *Viridans* 2(25%) | | CoNS 1(16.67%) | |
| *Enterococci* 11 (20.75%) | *Other Streptococci* 1 (20%) | *Staph aureus* 4 (13.8%) | *Viridans*3(17%) | *Viridans* 1(10%) | CoNS41(16.8%) | | CoNS2(25%) | | *Enterococci*1 (16.67%) | |
| *Viridans* 3 (5.7%) | GNR 1(20%) | *Viridans* 1(3.45%) | *Other Streptococci* 3 (17%) | CoNS 1(10%) | *Viridans* 16(6.9%) | | *Staph aureus* 1 (12.5%) | | *Other Streptococci* 1 (16.67%) | |
| Others 13 (24.5%) | | *Other Streptococci* 4 (13.8%) | CoNS2(11%) | GNR 2(20%) | Negative Cultures 12(5.2%) | | *Other Streptococci* 1 (12.5%) | | GNR 1(16.67%) | |
| Negative cultures 2 (3.8%) | | *HACEK* 1(3.45%) | | *Candida parapsilosis* 1 (10%) | [Information for causative organisms is available for 180 patients] | | | | Negative culture 1(16.67%) | |
| | | GNR 1(3.45%) Polymicrobial 1 (3.45%) | | | | | | | | |
| | | Negative cultures 5 (17.24%) Not available 3 (10.34%) | | | | | | | | |

**CoNS**: *Coagulase-negative Staphylococcus* species, **Staph aureus**: *Staphylococcus aureus*, **GNR**: Gram negative rod, **HACEK**:*Haemophilus*species, *Actinobacillus*, *Cardiobacterium hominis*, *Eikenellacorrodens* and *Kingella* species

**Polymicrobial:** Blood cultures of 1 patient was positive for *Enterococcus faecalis* and *coagulase-negative Staphylococci*

**Table 5. Clinical outcomes of post-TAVI infective endocarditis.**

| Amat-Santos | Puls | Latib | Olsen | Martinez-Selles | Regueiro | Seiffert | Spartera | Kosek | Gallouche | Doss |
|---|---|---|---|---|---|---|---|---|---|---|
| **MACE** | | | | | | | | | | |
| In-hospital Mortality | | | | | | | | | | |
| 25(47.2%) | | 13(44.8%) | 2(11%) | 2(20%) | 90(36%) | | | 1(14.3%) | 2(33.3%) | |
| Mortality at Follow-up | | | | | | | | | | |
| 13(24.5%) | 2(20%) | 5(17.2%) | 2(11%) | 3(30%) | 50(31.5%) | | 6(75%) | | | |
| Heart Failure | | | | | | | | | | |
| 36(67.9%) | 1(20%) | 9(31%) | | 3(30%) | 87(36.6%) | | | | | |
| Stroke | | | | | | | | | | |
| | | | | | 25(10.5%) | 0 | | | | |
| Major Bleeding | | | | | | | | | | |
| | | 2(11%) | | | 29(11.6%) | | | | | |
| Arrhythmias | | | | | | | | | | |
| | | | | 2(20%) | | | | | | |
| **NACE** | | | | | | | | | | |
| Septic Shock | | | | | | | | | | |
| | | | | 1(10%) | 66(27.7%) | | | | | |
| **Surgical intervention** | | | | | | | | | | |
| 4(7.5%) | 1(20%) | 3(10.3%) | 1(5.6%) | 2(20%) | 37(14.8%) | 1(3.2%) | | | | |
| **Valve-in-Valve Procedure** | | | | | | | | | | |
| 2(3.8%) | | 1(3.4%) | 3(17%) | | 3(1.2) | | | | | |

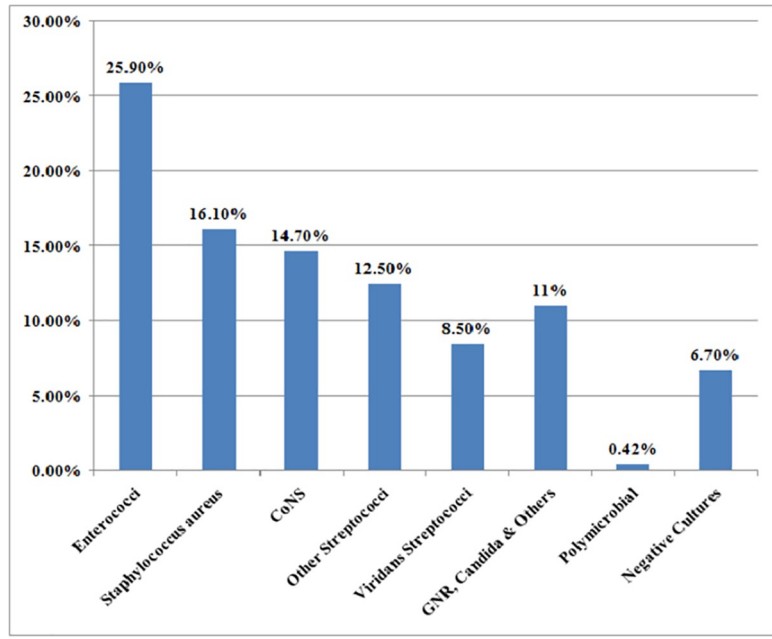

**Fig 5. Causative organisms of post-TAVI infective endocarditis.** The in-hospital mortality and mortality at follow-up were assessed in 7 studies with the mean incidence of 29.5% in-hospital mortality and 29.9% mortality at follow-up. The incidence of heart failure after post-TAVI infective endocarditis was reported in 5 studies making a cumulative incidence of 37.1%. Two studies determined the incidence of stroke with the stroke occurring in 10.5% and 0% post-TAVI infective endocarditis patients. Major bleeding was reported in 2 studies with an average of 11.3%. Only a single study by Martinez-Selles et al. reported arrhythmias in 20% cases. The septic shock occurred in 10% and 27.7% post-TAVI infective endocarditis patients according to 2 studies. The surgical intervention for the treatment of post-TAVI infective endocarditis was reported in 7 studies with the mean of 11.4%. Four studies revealed the valve-in-valve procedure performed in an average of 6.4% cases. The clinical outcomes of post-TAVI infective endocarditis are shown in Fig 6.

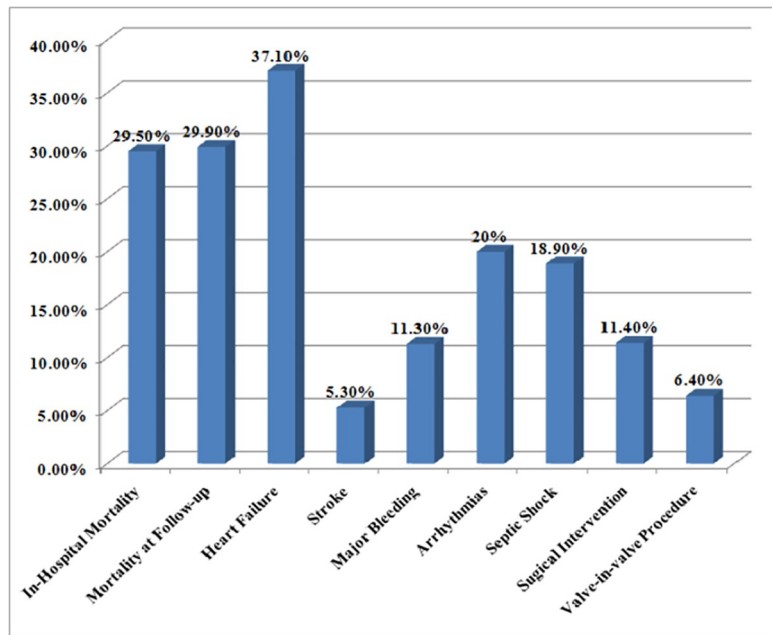

**Fig 6. Clinical outcomes in patients of post-TAVI infective endocarditis.** The incidence of infective endocarditis in the included studies with its causative organisms are summarized in Table 4.

after post-TAVI infective endocarditis[19]. The major limitation of these systematic reviews was that most of the included articles were case reports and case series which may have precluded the real evaluation of the post-TAVI infective endocarditis. Lastly, certain cases of post-TAVI infective endocarditis might not have been published leading to potential publication bias.

A retrospective multicenter study was conducted in 21 centers in America and Europe. A total of 7944 patients underwent TAVI out of which 53(0.67%) patients developed infective endocarditis. The mean follow-up time of patients was 1.1±1.2 years. Antibioticprophylaxis was given in all centers during the TAVI procedure. Cephalosporins, vancomycin and piperacillin/tazobactam were used in 14(67%), 6(28%) and 1(5%) centers respectively. In most of the centers, a single antibiotic dose was administered except 2 centers in which 2–3 doses were also given after TAVI. *Coagulase-negative Staphylococcus* specieswas the most causative organism isolated from 13(24.5%) cases.*Staphylococcusaureus*, *Enterococci* and *Viridans streptococci* were isolated from 11(20.8%), 11(20.8%) and 3(5.7%) patients, respectively. Infective

**Table 6. Antibiotics used for treatment of post-TAVI IE in 5 studies.**

| Study | Antibiotics | | | | |
|---|---|---|---|---|---|
| | β lactam drugs | Vancomycin | Gentamicin | Rifampicin | Daptomycin |
| Regueiro* | 164(75.6%) | 53(24.4%) | | | |
| Amat-Santos** | 21(39.6%) | 16(30.2%) | 20(37.7%) | 7(13.2%) | 5(9.4%) |
| Martinez-Selles | 9(90%) | | | | |
| Olsen | 13(72.2%) | 5(27.8%) | | | |
| Puls | 2(40%) | 2(40%) | | | |

*217 patients were treated with antibiotics.

** Some patients had received antibiotics in combination. That is why the percentage of antibiotics is not 100%.

endocarditis was caused by atypical microorganisms such as *Escherichia coli*, *Serratia*, *Acinetobacter*, *Candida lusitaniae*in 13(24.5%) patients. In 2(3.8%) patients, cultures were negative. The most common complication of IE was heart failure in 36(67.9%) patients. The other complications were acute kidney injury, septic shock, systemic embolism, stroke and persistent infection. The patients were managed with surgical explantation (7.5%) and valve-in-valve procedure (3.8%). In-hospital mortality was reported in 25(47.2%) and mortality at follow-up in 13(24.5%) patients[23].

Another study reported the incidence and clinical outcomes of infective endocarditis after TAVI retrospectively from 2008 to 2013. The mean follow-up time was 393 days. Prophylactic antibiotics were given in all patients. Out of 2572 TAVI patients, 29(1.13%) patients developed infective endocarditis. The most frequent pathogen was *Enterococci*present in 5(17.2%) patients followed by *coagulase-negative Staphylococcus* species in 4(13.8%) patients. Other organisms isolated were *Staphylococcus aureus* in 4(13.8%), *Streptococci* other than *Viridans* in 4(13.8%), *Viridans streptococci* in 1(3.4%), *HACEK* in 1(3.4%), gram negative rods in 1(3.4%) and polymicrobial in 1 patient. Cultures were negative in 5(17.2%) and not available in 3 (10.3%) patient. Heart failure occurred in one-third of the population. The outcomes of IE were in-hospital mortality in 13(44.8%) and mortality at follow-up in 5(17.2%) patients, surgery in 3(10.3%) and TAVR-in-TAVR in 1(3.4%) patient. The patients during follow-up died from the stroke (3), relapse of IE (1) and sepsis (1)[33].

A retrospective study was done on post-TAVI infective endocarditis from 2005 to 2015 including 47 centers in America and Europe. The follow-up time was 10.5 months. Among the 20006 patients who underwent TAVI, infective endocarditis occurred in 250(1.3%) patients. The data of causative pathogens was available for 180 patients. The most common organism was *Enterococci* found in 57(24.6%) cases followed by *Staphylococcus aureus* in 54(23.3%) patients. *Coagulase-negative Staphylococcus*speciesand *Viridans streptococci* were present in 41 (16.8%) and 16(6.9%) patients, respectively. Negative cultures were reported in 12(5.2%) cases. The complications of infective endocarditis were heart failure (36.6%), in-hospital mortality (36%), surgical intervention (14.8%), transcatheter valve-in-valve procedure (1.2%), pacemaker extraction (2.8%), recurrent IE (9.4%) and death during follow-up (31.5%)[28].

Another study reported 739 cases of IE from 26 Spanish hospitals. Ten (1.1%) cases of post-TAVI infective endocarditis occurred out of total 952 TAVI patients and 221(29.9%) after SAVR. The organisms isolated were as follows: *Enterococci* in 3(30%), *Viridans streptococci* in 1(10%), other *Streptococci* in 2(20%), *coagulase-negative Staphylococcus* species in 1(10%), *Salmonella enteritidis* in 1(10%), *Acinetobacter* in 1(10%) and *Candida parapsilosis* in 1(10%) patient. Three (30%) patients developed heart failure, 1(10%) underwent surgery, 1(10%) had a relapse of IE, 2(20%) patients died during hospitalization and 3(30%) during follow-up[22].

In Denmark, a study was done including 509 consecutive post-TAVI patients. Eighteen (3.5%) patients developed infective endocarditis during the follow-up of 1.4 years. Blood culture revealed *Enterococci* in 6(33%), *Staphylococcus aureus* in 4(22%), *Viridans streptococci* in 3 (17%), other *Streptococci* in 3(17%) and *coagulase-negativeStaphylococcus*speciesin 2 cases (11%). The outcomes of IE were in-hospital mortality in 2(11%), mortality at follow-up in 2 (11%), surgical intervention in 1(5.6%), a transcatheter valve-in-valve procedure in 3(17%) and pacemaker implantation in 4(22%)[27].

In another study, 180 consecutive post-TAVI patients were followed up to 319 days. Five (2.8%) cases of infective endocarditis occurred during that period. The following pathogens were isolated from the blood cultures of the patients: *Enterococci* in 2(40%), *Staphylococcus aureus* in 1(20%), other *Streptococcus* species in 1(20%) and gram negative rods in 1(20%) patient. One patient developed heart failure (20%), 1(20%) patient underwent surgery and 2 (40%) patients died after infective endocarditis[34].

In Grenoble Alpes University Hospital France, a retrospective study was done on 326 patients who underwent TAVI with the average follow-up period of 460 days. Out of 326 patients, 6 (1.8%) patients were diagnosed with infective endocarditis. Blood cultures were positive in 5 and negative in 1 patient. The pathogens responsible in 5 patients were *Staphylococcus aureus*, *coagulase-negative Staphylococcus* species, *Enterococci*, *Streptococcus* species and *Escherichia coli*. Two patients died after infective endocarditis, one 17 days and another 40 days after IE[35].

Kosek et al. retrospectively analyzed 7 patients with bicuspid aortic valve who underwent TAVI at the Institute of Cardiology, Poland. A total of 104 patients had TAVI at the institution and 7 of them had the bicuspid aortic valve. These patients were followed up to 12 months after TAVI. One patient developed infective endocarditis and died after 30 days of the procedure[36].

A study was carried out on 100 patients who underwent TAVI in Germany. The average follow-up duration was 3.8±2 years. The outcomes assessed were 30-day mortality, 5-year mortality, infective endocarditis, embolization, reintervention, stroke and valve thrombosis. None of the patients developed endocarditis[37].

Another study analyzed the outcomes of TAVI in patients with severe aortic regurgitation. Thirty one patients were included from 9 German centers. The follow-up time was 235 days. During follow-up, one of the patients had infective endocarditis and required surgical intervention (SAVR)[18].

Another retrospective study was performed in Italy in which 621 post-TAVI patients were followed for 402 days. Eight (6%) patients developed infective endocarditis. The microorganisms isolated were *Enterococci* in 2, *Viridans streptococci* in 2, *coagulase-negativeStaphylococcus*species in 2, *Staphylococcus aureus* in 1 and *Streptococcus* speciesin 1 patient. The mortality rate was 75% in patients[29].

## Conclusions

The incidence of post-TAVI infective endocarditis is low being 3.25% but it is associated with high mortality and complications. The most common complication is heart failure with a cumulative incidence of 37.1%. *Enterococci*are the most common causative organism isolated from 25.9% of cases followed by *Staphylococcus aureus*in 16.1% of cases. Appropriate measures should be taken to prevent infective endocarditis in post-TAVI patients including adequate antibiotics prophylaxis directed specifically against these organisms.

## Limitations of the study

- The meta-analysis could not be done as there was no control/comparison group in the included studies.

- The outcomes of post-TAVI IE have not been integrated with the impact of different treatment options as none of the included studies correlated the outcomes with the treatment modality.

- The review does not improve the discussion on IE prophylaxis, although the rationale sets this topic as a central objective. This is because the data on IE prophylaxis is not detailed in all the included studies.

## Recommendations of the study

- Meticulous aseptic measures should be reinforced with special focus on sterilization and disinfection in the catheterization laboratory. The provision of laminar air flow has an added advantage.

- Antibiotic prophylaxis should be given in TAVI patients. As *Enterococci* are the most common organism causing post-TAVI infective endocarditis, the prophylactic antibiotics should be given directed against this organism. The traditional use of cephalosporins should be reconsidered as *Enterococci* are intrinsically resistant to all cephalosporins.

## Supporting information

**S1 Annexure. Search strategy.**
(DOCX)

**S2 Annexure. Search strategy on pubmed.**
(DOCX)

**S3 Annexure. PRISMA checklist.**
(DOCX)

## Author Contributions

**Conceptualization:** Adnan Khan.

**Data curation:** Adnan Khan, Aqsa Aslam, Khawar Naeem Satti, Sana Ashiq.

**Investigation:** Adnan Khan, Khawar Naeem Satti.

**Methodology:** Adnan Khan, Aqsa Aslam, Khawar Naeem Satti, Sana Ashiq.

**Project administration:** Adnan Khan.

**Resources:** Adnan Khan.

**Supervision:** Adnan Khan.

**Validation:** Aqsa Aslam, Khawar Naeem Satti, Sana Ashiq.

**Visualization:** Aqsa Aslam, Khawar Naeem Satti, Sana Ashiq.

**Writing – original draft:** Aqsa Aslam.

**Writing – review & editing:** Aqsa Aslam, Khawar Naeem Satti, Sana Ashiq.

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
