## [Decision Letter · Decision Letter 0]

24 Jul 2019

PONE-D-19-17992

Infective Endocarditis Post-Transcatheter Aortic Valve Implantation (TAVI), Microbiological Profile and Clinical Outcomes: A Systematic Review

PLOS ONE

Dear Dr. Adnan Khan,

Thank you for submitting your manuscript to PLOS ONE. After careful consideration, we feel that it has merit but does not fully meet PLOS ONE’s publication criteria as it currently stands. Therefore, we invite you to submit a revised version of the manuscript that addresses the points raised during the review process.

ACADEMIC EDITOR: Although it is of interest, the reviewers have raised a number of points which we believe major modifications are necessary to improve the manuscript, taking into account the reviewers' remarks

We would appreciate receiving your revised manuscript by Sep 07 2019 11:59PM. To enhance the reproducibility of your results, we recommend that if applicable you deposit your laboratory protocols in protocols.io, where a protocol can be assigned its own identifier (DOI) such that it can be cited independently in the future. For instructions see: http://journals.plos.org/plosone/s/submission-guidelines#loc-laboratory-protocols

We look forward to receiving your revised manuscript.

Kind regards,

Wisit Cheungpasitporn, MD, FACP

University of Mississippi Medical Center

Twitter: @wisit661 Email: wcheungpasitporn@gmail.com 

Academic Editor

PLOS ONE

Journal Requirements:

3. Please include in your methods section both the dates included in your search and the dates during which the electronic search was performed.

4. We note that Table 2 in your submission has been previously published. All PLOS content is published under the Creative Commons Attribution License (CC BY 4.0), which means that the manuscript, images, and Supporting Information files will be freely available online, and any third party is permitted to access, download, copy, distribute, and use these materials in any way, even commercially, with proper attribution. For more information, see our copyright guidelines: http://journals.plos.org/plosone/s/licenses-and-copyright.

We require you to either (1) present written permission from the copyright holder to publish this table specifically under the CC BY 4.0 license, or (2) remove the table from your submission:

1.         You may seek permission from the original copyright holder of table 2 to publish the content specifically under the CC BY 4.0 license.

In the table title of the copyrighted table, please include the following text: “Reprinted from [ref] under a CC BY license, with permission from [name of publisher], original copyright [original copyright year].”

2.    If you are unable to obtain permission from the original copyright holder to publish this table under the CC BY 4.0 license or if the copyright holder’s requirements are incompatible with the CC BY 4.0 license, please either i) remove the figure or ii) supply a replacement figure that complies with the CC BY 4.0 license. Please check copyright information on all replacement figures and update the figure caption with source information. If applicable, please specify in the figure caption text when a figure is similar but not identical to the original image and is therefore for illustrative purposes only.

4. Please include a caption for figure 3.

Reviewers' comments:

Reviewer's Responses to Questions

**Comments to the Author**

1. Is the manuscript technically sound, and do the data support the conclusions?

Reviewer #1: Yes

Reviewer #2: Partly

Reviewer #3: Partly

Reviewer #4: Yes

2. Has the statistical analysis been performed appropriately and rigorously? 

Reviewer #1: Yes

Reviewer #2: No

Reviewer #3: N/A

Reviewer #4: N/A

3. Have the authors made all data underlying the findings in their manuscript fully available?

Reviewer #1: Yes

Reviewer #2: Yes

Reviewer #3: No

Reviewer #4: Yes

4. Is the manuscript presented in an intelligible fashion and written in standard English?

Reviewer #1: Yes

Reviewer #2: No

Reviewer #3: Yes

Reviewer #4: Yes

5. Review Comments to the Author

Reviewer #1: This article showed a metanalysis based on Infective Endocarditis Post-Transcatheter Aortic Valve Implantation (TAVI). Eleven articles were included in the systematic review. The authors wanted to study the incidence of infective endocarditis in post-TAVI as well as its microbiological profile and clinical outcomes.

This metanalysis was very interesting and it could be very useful to improve the knowledge in this field.

However, there were some points to discuss:

Major issues

• (Page 9, line 276) The authors should improve the data synthesis. They must declare if and how many studies were not considered and, eventually, which criteria did not respect for the inclusion in the metanalysis.

• Why was not the antibiotic therapy during the Infective Endocarditis Post-Transcatheter Aortic Valve Implantation specified?

• Were there different outcomes based on the different aetiology? In the metanalysis should be explain why this cannot be done. This is a critical and important point.

Minor issues

• Name of the bacteria:

o Viridians streptococci – probably better Viridans streptococci;

o for example: Staphylococcus species – probably better Staphylococcus species or spp. (species not in Italic style). The same was for other bacteria cited in the Metanalysis.

• (Page 3, line 82) Reference missed.

• (Page 3, line 83) The authors wrote “Ambrosioni et al. reported that infective endocarditis is responsible for 25% of deaths”. It was not clear which kind of death the authors mean.

• (page 9, line 278) “the follow-up duration of at least 6 months. 0”. Delete the part after the full stop.

• (Page 12, table 5) In the Amat-Santos, Latib, Gallouche studies, the sum of the percentages of the microorganism are not 100%.

• (Page 12, table 5) It could be better if the authors split the table 5 into 2 different tables (one with the incidence and the ethology, and one with the outcomes.

• (Page 13, line 331-332) The sum of the percentages of the complications are not 100%. It is better to explain if there were other complications.

• (Page 13, line 335, and in other lines) Enterococcus – Enterococcus spp. or Enterococci

• (Page 14, line 347) Staphylococus – It is better Staphylococcus

• (Page 14, line 351-353) It is not really clear the meaning of the sentence. Maybe, check the sum of the percentage.

Reviewer #2: This systematic review evaluated the incidence of infective endocarditis after transcatheter aortic valve implantation (TAVI).The incidence from 11 studies was 0 to 14.3% and the commonest organism was enterococcus.This was associated with a high mortality.

The study is of value as this is an emerging issue as transvalvular interventions evolve.

Rationale：sentences are made without references

Method: the analysis plan is not described

Results: standardized plots not reported，Results are mainly descriptive. Bias and validity was not assessed.

There are far too many figures. The value of the photography is unclear.

Discussion: It is recommended that the discussion be amended to begin with a statement of the principle findings as listed as the primary and secondary aims.Then a summary in 3-5 paragraphs as to how the findings of this systematic review are similar or different to other findings in the literature and possible reasons why this is so. Also discuss the limitations of the findings.

In general, abbreviations need to be pre- defined prior to use.

Reviewer #3: Khan et al. present a systematic review on Infective Endocarditis (IE) occurring after Transcatheter Aortic Valve Implantation (TAVI). Analyzing 11 articles, they focused on the incidence, microbiological profiles and clinical outcome (i.e. mortality, stroke, bleedings, heart failure, septic shock, and arrhythmias) of IE. According to their conclusions post-TAVI IE is mostly linked to Enterococcus, while native valve IE is often associated to S. Aureus or Streptococcus; the mortality rate was about 30%; the

most common complication was heart failure.

Overall, the manuscript is not really well-written due to some grammar mistakes and lack of text fluidity, while the length is appropriate for a systematic review. Tables are globally well constructed and and figures appear explicative, but there is space for refinement in small layout imperfections.

I have the following points of criticism:

Major:

• The manuscript is not really well-written due to some English mistakes and no fluidity among different statements;

• Introduction explains TAVI and IE, but no identification of a central purpose and no explanation of what the study adds to the knowledge are clearly available;

• Discussion mentions Pulmonary Valve Replacement (PVR), despite not being even mentioned in the Introduction; moreover, no studies on PVR were included in the systematic review;

• Authors state that no other reviews or meta-analysis are available focusing on this topic, despite at least two well-written systematic reviews from Ando et al. and Amat-Santos et al. have been recently published; it might be interesting to show the limitations of these articles, trying to overcome them;

• It is mandatory to explain differences and similarities between native valve-IE and TAVI-IE; in particular, concentrating on different timing and diagnostic management;

• The review does not improve the discussion on IE prophylaxis, although the Rationale sets this topic as a central objective;

• It is necessary to explain why case report and more over case series were excluded from this systematic review;

• The Discussion sometimes merely mentions the results without making them homogeneous in light of literature evidences; there is a need for original arguments to make the Discussion more appealing.

• Authors do not include some relevant studies in the systematic review (i.e. Onsea et. al., Buellesfeld et al., Barbanti et al.).

• Since Authors have analysed only two articles focusing on stroke, mean of stroke incidence may be influenced by extreme values (0% vs. 10.3%);

• IE outcomes are analyzed and summarized, but it would be necessary to integrate them with the impact of different treatment options (i.e. antibiotics therapy, surgery) to be more specific;

Minor:

• Scientific English and grammar should be globally improved;

• All measures should be accompanied by a dispersion index;

• Figure 3 reports a wrong green underscore;

• Figure 4 has some troubles in the layout;

• Figure 5 has some needless dots in a caption;

• Line 278 contains a redundant “0”;

• Table 5 at “valve-in-valve procedure” line miss a “%”; in addition, some words appear in italic as a mistake

Reviewer #4: Please also include timeline of the literature search in the method section of the abstract.

Who are two independent investigators?

Figure 3 (search flowchart), suggest to use PRISMA 2009 Flow Diagram platform

Current quality of all figures are not acceptable. They are very difficult to evaluate. Will need higher quality for all of them.

Search terms in Medline and Embase are different. Please attach syntax used in each database as supplementary.

Please make the data for this review publicly available, possibly through the Open Science Framework (osf.io). Items to include: list of excluded studies, etc. Making data publicly available will promote the reproducibility of the review and is best practices for systematic reviews.

6. PLOS authors have the option to publish the peer review history of their article (what does this mean?). If published, this will include your full peer review and any attached files.

Reviewer #1: No

Reviewer #2: No

Reviewer #3: No

Reviewer #4: No

---

## [Author Response · Author response to Decision Letter 0]

4 Sep 2019

The manuscript has been revised as suggested by the reviewers. Each point has been explained in this document. The marked and unmarked copies of the revised manuscript have been submitted.

Journal Requirements:

• The manuscript meets PLOS ONE's style requirements, including those for file naming. 

• The Supporting Information files have been included at the end of manuscript. 

• The dates during which the electronic search was performed have been mentioned.

• Table 2 has been omitted from the manuscript.

• The caption for figure 3 has been included.

Answers to the Questions

Reviewer's Responses to Questions

Comments to the Author

4. Is the manuscript technically sound, and do the data support the conclusions?

Reviewer #1: Yes

Reviewer #2: Partly

Reviewer #3: Partly

Reviewer #4: Yes

The manuscript is technically sound and the data support the conclusions. The study is a systematic review which included all the relevant articles on post-TAVI infective endocarditis till October 2018. The literature search was systematically conducted on 3 electronic databases. Medline (Pubmed), Embase and the Cochrane Central Register of Controlled Trials. The conclusion of the study correlates with the data and is drawn from the results of the data analyzed.

2. Has the statistical analysis been performed appropriately and rigorously? 

Reviewer #1: Yes

Reviewer #2: No

Reviewer #3: N/A

Reviewer #4: N/A

The data is analyzed using mean, frequency & percentages and presented as tables & bar graphs. The meta-analysis could not be done as there was no comparison/control group.

3. Have the authors made all data underlying the findings in their manuscript fully available?

Reviewer #1: Yes

Reviewer #2: Yes

Reviewer #3: No

Reviewer #4: Yes

The data of the manuscript is provided in the main text and supporting information. The data of systematic search is shown in data synthesis & PRISMA flow diagram. The important parameters of included students are tabulated in the results. The search strategy, Pubmed syntax and PRISMA checklist are provided in supporting information.________________________________________

4. Is the manuscript presented in an intelligible fashion and written in standard English?

Reviewer #1: Yes

Reviewer #2: No

Reviewer #3: Yes

Reviewer #4: Yes

The manuscript is written in standard English which is clear and correct. Any grammatical mistake has been corrected in the revised manuscript.________________________________________

5. Review Comments to the Author

Reviewer #1: 

Major issues

• (Page 9, line 276) The data synthesis has been improved. The studies considered and the criteria which did not respect for the inclusion in the meta-analysis are mentioned.

• The antibiotic therapy during post-transcatheter aortic valve implantation infective endocarditis is not specified because the data on antibiotic therapy was not available in the included studies.

• The meta-analysis could not be done as there was no control/comparison group in the included studies. 

Minor issues

• Name of the bacteria:

The name of Viridians streptococci has been changed to Viridans streptococci

Staphylococcus species: Species is written in non-italic style 

• (Page 3, line 82) The Reference number is 1.

• (Page 3, line 83) The line has been rephrased.

• (page 9, line 278) The redundant 0 after the full stop has been deleted.

• (Page 12, table 5) The sum of the percentages of the microorganisms are corrected to make 100% in the Amat-Santos, Latib, Gallouche studies.

• (Page 12, table 5) Table 5 has been split into 2 different tables (one with the incidence & the etiology and one with the outcomes).

• (Page 13, line 331-332) In the relevant article, congestive heart failure is the presenting feature of post-TAVI IE. That is why the sum of the percentages are not 100%. The details have been mentioned in the manuscript.

• (Page 13, line 335, and in other lines) Enterococcus has been changed to Enterococci.

• (Page 14, line 347) The spelling of Staphylococcus has been corrected.

• (Page 14, line 351-353) The sentence has been rephrased.

Reviewer #2: 

• Rationale: The references of sentences have been mentioned in the rationale.

• Method: The analysis plan has been described.

• Results: The results are expressed as tables and presented as Box plots. The risk of Bias is calculated and shown in table 3.

• The figures have been uploaded in the format (TIF and PACE) as suggested by the journal. 

• Discussion: The discussion has been amended with the principle findings of the study. 

• Limitations of the findings: The limitations have been added.

• Abbreviations have been pre-defined prior to use.

Reviewer #3: 

Major:

• The manuscript has been checked for English mistakes. 

• Introduction: TAVI and IE have been explained first with further details of post-TAVI IE. The central purpose and explanation of what the study adds to the knowledge are clearly mentioned in the rationale of the study.

• This systematic review is conducted on post-TAVI IE. The reference of a systematic review by Amat-Santos et al. has been mentioned in the discussion. Amat-Santos et al. conducted a systematic review on infective endocarditis after transcatheter aortic as well as pulmonary valve replacement.

• It has been stated in the introduction of the manuscript that the data on post-TAVI IE is limited and most of the available studies are case reports & case series. The systematic review by Amat-Santos et al. has been explained in the discussion. Most of the included studies in this review were also case reports and case series. The study by Ando et al. is a meta-analysis that compared the incidence of infective endocarditis following TAVI versus surgical aortic valve replacement. The study is published in 2019 whereas the literature search for our systematic review was conducted till October 2018.

Amat-Santos IJ, Ribeiro HB, Urena M, Allende R, Houde C, Bedard E, et al. Prosthetic valve endocarditis after transcatheter valve replacement: a systematic review. JACC Cardiovasc Interv. 2015 Feb; 8(2):334–46. doi: 10.1016/j.jcin.2014.09.013.

Ando T, Ashraf S, Villablanca PA, Telila TA,Takagi H, Grines CL, et al. Meta-analysis comparing the incidence of infective endocarditis following after transcatheter aortic valve implantation versus surgical aortic valve replacement. Am J Cardiol. 2019 Mar 1; 123(5):827-32. doi: 10.1016/j.amjcard.2018.11.031.

• The differences and similarities between native valve-IE and TAVI-IE; in particular, concentrating on different timing and diagnostic management have been mentioned in the introduction.

• The review does not improve the discussion on IE prophylaxis, although the Rationale sets this topic as a central objective. This is because the data on IE prophylaxis is not detailed in the included studies.

• The case reports and case series were excluded from this systematic review. This is because most of the included studies in the two systematic reviews (available in the literature) on post-TAVI IE were case reports & case series. The major limitation of these reviews was that it may have precluded the real evaluation of the post-TAVI infective endocarditis and certain cases of post-TAVI infective endocarditis might not have been published leading to potential publication bias. The details of these two systematic reviews by Amat-Santos et al. [26] & Eisen et al. [19] and their limitations have been mentioned in the discussion.

• The results of the study have been explained in the discussion along with comparison with the results of two systematic reviews available. The results of our systematic review cannot be compared with the findings of the included studies. The details of the included studies have been mentioned in the discussion. 

• The studies by Onsea et. al., Buellesfeld et al. and Barbanti et al. were not included in the systematic review because these studies do not meet the inclusion criteria. The details of these studies with their references are given below:

In a study by Onsea et al., a total of 73 TAVI patients were analyzed for the development of complications. Eleven patients developed infections but none of them was diagnosed with infective endocarditis [1].

In a study by Barbanti et al., the patients who underwent TAVI were categorized into two groups: those with Society of Thoracic Surgeons (STS) score ≤7% and patients with a score >7%. The outcomes of TAVI were compared between these 2 groups. The results showed that the patients with STS ≤7% had lower rates of all-cause and cardiovascular mortality at 3 years after transcatheter aortic valve implantation.

In another study by Barbanti et al., the outcomes of TAVI were compared in patients discharged within 72 hours and after 3 days of the procedure. The study showed that discharge within 72 hours after TAVI is feasible and does not seem to jeopardise the early safety of the procedure [3].

Buellesfeld et al. evaluated the outcomes and predictors of success of procedure in patients who underwent TAVI at two institutions in German & Switzerland. The reported rates of in-hospital mortality, myocardial infarction and stroke were 11.9%, 1.8% and 3.6%, respectively.

1. Onsea K, Agostoni P, Voskuil M, Samim M, Stella PR. Infective complications after transcatheter aortic valve implantation: results from a single centre. Neth Heart J. 2012; 20:360–4. doi: 10.1007/s12471-012-0303-9.

2. Barbanti M, Schiltgen M, Verdoliva S, Bosmans J, Bleiziffer S, Gerckens U, et al. Three-year outcomes of transcatheter aortic valve implantation in patients with varying levels of surgical risk (from the CoreValve ADVANCE Study). Am J Cardiol. 2016; 117(5):820-7.

3. Barbanti M, Capranzano P, Ohno Y, Attizzani GF, Gulino S, Imme S, et al. Early discharge after transfemoral transcatheter aortic valve implantation. Heart. 2015; 101:1435-6. doi: 10.1136/heartjnl-2015-308415.

4. Buellesfeld L, Wenaweser P, Gerckens U, Mueller R, Sauren B, Latsios G, et al. Transcatheter aortic valve implantation: predictors of procedural success—the Siegburg–Bern experience. Eur Heart J. 2010; 31:984–91. doi:10.1093/eurheartj/ehp570.

• There were only two articles reporting on stroke after post-TAVI IE. The mean of stroke incidence may be influenced by extreme values (0% vs. 10.3%). The mean of stroke has been omitted from the manuscript.

• Infective endocarditis outcomes have not been integrated with the impact of different treatment options (i.e. antibiotics therapy, surgery). Because none of the included study correlated the outcomes with the treatment modality. 

Minor:

• Scientific English and grammar have been improved.

• The green underscore has been omitted in figure 3.

• Corrected copy of figure 4 has been submitted. 

• Corrected copy of figure 5 has been submitted.

• Redundant “0” in line 278 has been omitted.

• The sign of % has been added after valve-in-valve procedure” in Table 5. In addition, some words which appear in italic as a mistake are corrected.

Reviewer #4: 

• The timeline of the literature search has been mentioned in the method section of the abstract.

• The first and second authors were the principal investigators who performed the literature search. It has been mentioned under the heading of data management, selection and data collection. 

• Figure 3 (PRISMA flow diagram) has been made by using PRISMA 2009 Flow Diagram platform.

• The figures have been uploaded in the format suggested by the journal. 

• The syntax used in Pubmed is attached in the supplementary file.

---

## [Decision Letter · Decision Letter 1]

18 Sep 2019

PONE-D-19-17992R1

Infective Endocarditis Post-Transcatheter Aortic Valve Implantation (TAVI), Microbiological Profile and Clinical Outcomes: A Systematic Review

PLOS ONE

Dear Adnan Khan,

Thank you for submitting your manuscript to PLOS ONE. After careful consideration, we feel that it has merit but does not fully meet PLOS ONE’s publication criteria as it currently stands. Therefore, we invite you to submit a revised version of the manuscript that addresses the points raised during the review process.

ACADEMIC EDITOR: Our expert reviewer(s) have recommended some major revisions to your manuscript. Therefore, I invite you to respond to the reviewer(s)' comments as below and revise your manuscript.

We would appreciate receiving your revised manuscript by Nov 02 2019 11:59PM. To enhance the reproducibility of your results, we recommend that if applicable you deposit your laboratory protocols in protocols.io, where a protocol can be assigned its own identifier (DOI) such that it can be cited independently in the future. For instructions see: http://journals.plos.org/plosone/s/submission-guidelines#loc-laboratory-protocols

We look forward to receiving your revised manuscript.

Kind regards,

Wisit Cheungpasitporn, MD, FACP

University of Mississippi Medical Center

Twitter: @wisit661 Email: wcheungpasitporn@gmail.com 

Academic Editor

PLOS ONE

Reviewers' comments:

Reviewer's Responses to Questions

**Comments to the Author**

1. If the authors have adequately addressed your comments raised in a previous round of review and you feel that this manuscript is now acceptable for publication, you may indicate that here to bypass the “Comments to the Author” section, enter your conflict of interest statement in the “Confidential to Editor” section, and submit your "Accept" recommendation.

Reviewer #1: (No Response)

Reviewer #2: All comments have been addressed

Reviewer #3: All comments have been addressed

Reviewer #4: All comments have been addressed

2. Is the manuscript technically sound, and do the data support the conclusions?

Reviewer #1: Partly

Reviewer #2: Yes

Reviewer #3: Yes

Reviewer #4: Partly

3. Has the statistical analysis been performed appropriately and rigorously? 

Reviewer #1: Yes

Reviewer #2: Yes

Reviewer #3: Yes

Reviewer #4: Yes

4. Have the authors made all data underlying the findings in their manuscript fully available?

Reviewer #1: Yes

Reviewer #2: Yes

Reviewer #3: Yes

Reviewer #4: Yes

5. Is the manuscript presented in an intelligible fashion and written in standard English?

Reviewer #1: Yes

Reviewer #2: Yes

Reviewer #3: Yes

Reviewer #4: Yes

6. Review Comments to the Author

Reviewer #1: I appreciate the suggested modification of the text, but I still have some doubts.

In particular the antibiotic therapy and the clinical characteristics of the cohort are not specified. For example, the presence of septic shock at the beginning, the developing of Multi-organ failure and Acute kid-ney failure have a deep impact on the mortality and the morbidity. The correct therapy, the empiric strategies, the fast changing of the therapy based on a microbial isolation and the timing of the treatment could change the clinical response and the presence of cardiological-releted complications.

I do not think this last version of the paper should be published, but a further integration with data about patient’s characteristics and therapeutic strategies from papers that report it, could give a more complete view and justify final conclusions on cardiological outcomes.

Reviewer #2: Suggest limitations be incorporated in discussion and the comments raised by reviewers have been addressed

Reviewer #3: Khan et al. have resubmitted a systematic review on Infective Endocarditis (IE) occurring after Transcatheter Aortic Valve Implantation (TAVI). Analyzing 11 articles, they focused on the incidence, microbiological profiles and clinical outcome (i.e. mortality, stroke, bleedings, heart failure, septic shock, and arrhythmias) of IE. According to their conclusions post-TAVI IE is mostly linked to Enterococcus, while native valve IE is often associated to S. Aureus or Streptococcus; the mortality rate was about 30%; the most common complication was heart failure.

Overall, after the Authors' revision the manuscript appear quite well-written with better text fluidity. The length is appropariate for a systematic review. Tables are globally well made and figures are explicative.

I would propose only the improvement of the following points of criticism:

Major:

• A rigorous definition of septic shock would be necessary;

Minor:

• All abbreviations could be better explained and revisited due to several futile repetitions;

• Line 418 contains a sentence with several grammar errors.

Reviewer #4: The investigators should obtain more information on patient’s characteristics, infection data and treatment from each included study and take it into consideration for additional analyses.

7. PLOS authors have the option to publish the peer review history of their article (what does this mean?). If published, this will include your full peer review and any attached files.

Reviewer #1: No

Reviewer #2: No

Reviewer #3: No

Reviewer #4: No

---

## [Author Response · Author response to Decision Letter 1]

2 Oct 2019

Reviewer 1:

The patients of post-TAVI IE were diagnosed according to the Modified Duke’s criteria. The data on antibiotic prophylaxis is detailed in 2 studies by Regueiro et al. and Amat-Santos et al. and it has been mentioned in the Results. The antibiotic therapy for post-TAVI IE has also been included in the manuscript. Recommendations have been added based on the conclusion of the manuscript.

Reviewer 3: 

• The definition of sepsis and septic shock are included in the manuscript in study outcomes (Methodology).

• All abbreviations are first preceded by complete word. After that, only abbreviations are written in the manuscript.

• The grammatical mistakes of the line no. 418 (now line no. 457) have been corrected.

Reviewer 4: 

The antibiotics used for treating post-TAVI IE are given in 5 studies. These details have been added in the results. The patients undergoing surgical treatment (valve explantation and valve-in-valve procedure) have already been included in the results. Due to the heterogeneity of data in the included studies, further analysis of the variables cannot be done.

---

## [Decision Letter · Decision Letter 2]

29 Oct 2019

Infective Endocarditis Post-Transcatheter Aortic Valve Implantation (TAVI), Microbiological Profile and Clinical Outcomes: A Systematic Review

PONE-D-19-17992R2

Dear Dr. Adnan Khan,

We are pleased to inform you that your manuscript has been judged scientifically suitable for publication and will be formally accepted for publication once it complies with all outstanding technical requirements.

With kind regards,

Wisit Cheungpasitporn, MD, FACP

University of Mississippi Medical Center

Twitter: @wisit661 Email: wcheungpasitporn@gmail.com 

Academic Editor

PLOS ONE

Additional Editor Comments:

I want to commend the authors on their superb efforts to revise the manuscript according to all reviewers’ suggestions. The quality of the manuscript has improved substantially.

Reviewers' comments:

Reviewer's Responses to Questions

**Comments to the Author**

1. If the authors have adequately addressed your comments raised in a previous round of review and you feel that this manuscript is now acceptable for publication, you may indicate that here to bypass the “Comments to the Author” section, enter your conflict of interest statement in the “Confidential to Editor” section, and submit your "Accept" recommendation.

Reviewer #1: All comments have been addressed

Reviewer #2: All comments have been addressed

Reviewer #3: All comments have been addressed

Reviewer #4: All comments have been addressed

2. Is the manuscript technically sound, and do the data support the conclusions?

Reviewer #1: Yes

Reviewer #2: Yes

Reviewer #3: Yes

Reviewer #4: Yes

3. Has the statistical analysis been performed appropriately and rigorously? 

Reviewer #1: Yes

Reviewer #2: I Don't Know

Reviewer #3: Yes

Reviewer #4: Yes

4. Have the authors made all data underlying the findings in their manuscript fully available?

Reviewer #1: Yes

Reviewer #2: No

Reviewer #3: (No Response)

Reviewer #4: Yes

5. Is the manuscript presented in an intelligible fashion and written in standard English?

Reviewer #1: Yes

Reviewer #2: Yes

Reviewer #3: Yes

Reviewer #4: Yes

6. Review Comments to the Author

Reviewer #1: Even if I understand the difficulty of finding all the data that I recommended, I think this last version is more complete and I appreciate the modification of the text introduced.

Infective Endocarditis Post-Transcatheter Aortic Valve Implantation is not so known and knowledge of the possible infective complications that could intercourse, the microorganism implicated and the outcomes instead are important data in order to choose the better treatment that can be given to the patient.

Finally, with this last version this work is deeply improved and now I think it could be published.

Reviewer #2: Overall all comments have been addressed as highlighted by each of the reviewers. There are no further outstanding issues.

Reviewer #3: (No Response)

Reviewer #4: All my concerns have been fully elucidated, missing sections and analyses have been completed. Finally, comprehension errors have been corrected. Good work!

7. PLOS authors have the option to publish the peer review history of their article (what does this mean?). If published, this will include your full peer review and any attached files.

Reviewer #1: No

Reviewer #2: No

Reviewer #3: No

Reviewer #4: No

---

## [Editor Report · Acceptance letter]

18 Dec 2019

PONE-D-19-17992R2 

Infective Endocarditis Post-Transcatheter Aortic Valve Implantation (TAVI), Microbiological Profile and Clinical Outcomes: A Systematic Review 

Dear Dr. Khan:

I am pleased to inform you that your manuscript has been deemed suitable for publication in PLOS ONE. Congratulations! Your manuscript is now with our production department. 

With kind regards,

on behalf of

Dr. Wisit Cheungpasitporn 

Academic Editor

PLOS ONE